# Here for My Peer: The Future of First Responder Mental Health

**DOI:** 10.3390/ijerph182111097

**Published:** 2021-10-22

**Authors:** Kristin A. Horan, Madeline Marks, Jessica Ruiz, Clint Bowers, Annelise Cunningham

**Affiliations:** 1University of Central Florida, Orlando, FL 32816, USA; jruiz2017@knights.ucf.edu (J.R.); Clint.Bowers@ucf.edu (C.B.); a.cunningham@knights.ucf.edu (A.C.); 2University of Maryland School of Medicine, Baltimore, MD 21201, USA; Madeline.Marks@ucf.edu

**Keywords:** peer support, mental health, first responders, process evaluation

## Abstract

Workplace interventions that leverage social tactics to improve health and well-being are becoming more common. As an example, peer mental health support interventions aim to reduce stigma and promote treatment seeking in first responder populations. Given the social nature of these interventions, it is important to consider how the preexisting social context influences intervention outcomes. A peer mental health support intervention was delivered among first responders, and self-efficacy and intention to have supportive peer conversations were measured pre-and post-intervention. Trust in peers was measured prior to the intervention. Results suggest a floor effect may exist for self-efficacy, in which a foundational level of trust and pre-intervention self-efficacy may be needed to maximize intervention effectiveness. As the future of work brings complex safety and health challenges, collaborative solutions that engage multiple stakeholders (employees, their peers, and their organization) will be needed. This study suggests that more frequent attention to pre-existing intervention context, particularly social context in peer-focused intervention, will enhance intervention outcomes.

## 1. Introduction

First responder professionals face increased job hazards, injuries, and fatalities on the job [1]. In addition to the dangers faced on the job, there are hidden consequences. A recent study found that firefighters are more likely to die of suicide than die on-the-job (even considering underreporting of first responder suicides) [2]. Focusing solely on the physical risks of the job is a disservice to these professionals. Therefore, a dual focus on the physical and mental health of first responders is imperative. 

Barriers exist that often prevent first responders from seeking mental health treatment [3]. One such barrier is social stigma related to treatment-seeking [4], which is driven by a high value placed on mental toughness [5]. To reduce social stigma barriers peer-directed interventions exist that target the social environment to encourage treatment-seeking behavior [6]. However, little is known about how the existing social environment influences an intervention that targets peers as a mechanism of change. Trust in workplace social relationships may influence the effectiveness of an intervention that depends on social actors to ultimately influence the target behavior. The present study will examine both outcomes and process for a peer mental health support intervention for first responders. The process evaluation component will focus on the influence of both pre-existing individual differences (self-efficacy) and pre-existing social context (levels of interpersonal trust in the workplace) on intervention outcomes. In doing so, we address calls for more research on how pre-existing context may influence intervention outcomes [7] and provide insight into factors that may help or hinder valuable intervention efforts aimed as first responder mental health. 

## 2. Interventions in the Context of the Future of Work

A consideration of pre-existing intervention context, particularly in the context of a first responder mental health intervention, is timely as researchers and practitioners respond to the changing nature of work. As the Future of Work Task Force points out, more research is needed on how organizational design can be used to address work stress and work-life issues [8]. Peer-directed interventions represent aspects of organizational design in that they leverage both formal and informal support systems to promote employee health and safety (including informal, naturally occurring supportive relationships, formal programs complimenting traditional mental health services such as employee assistance programs (EAPs), and peers employed in a dedicated peer-support role) [9]. These programs also integrate work and nonwork domains as they address a wide range of behavioral health issues affecting first responders [10], which can emerge on the job, at home, or both [11]. As work-life boundaries become increasingly blurred, organizational design programming that addresses the interplay between these domains becomes more important [8]. 

Additionally, the Future of Work Task Force also asserts that as workplaces and work stressors evolve, collaborative solutions that engage both leadership and employees become more important [8]. Borrowing from this logic that safety and health is not the employee’s sole responsibility, we extend the general idea of collaborative solutions to also include work peers as catalysts of change. In doing so, we examine the feasibility and efficacy of including the work peer as a key stakeholder in protecting and promoting employee quality of work life. Finally, the Future of Work Task Force points out that employee quality of work life and organizational ethical responsibility is perceived as increasingly intertwined. They assert that qualities of workplace relationships, including trust, will grow in their importance in organizational design to support employee health [8], meaning that employees may or may not trust their employer to fulfill their ethical responsibility to promote safe and healthy work. This paper examines trust in work peers as social context that influences outcomes in a peer-focused workplace intervention. Although such an approach applies to a wide range of work populations, work stressors, or health and safety outcomes, in this instance, we train work peers to protect and promote employee mental health in first responder populations. 

## 3. Mental Health & Suicide among First Responders

Interest in the psychological and physical effects of job-related stress for first responders is growing, given that first responders have high rates of on-the-job exposure to traumatic events due to the nature of their job [12]. Critical incidents refer to events that pose a significant risk for physical or psychological harm [13]. For many employees in the general population, the experience of a critical incident would be highly unusual, for first responders, they are routine [3]. For example, up to 60% of police officers witnessed or were involved in five or more critical incidents on the job during the past year, with 75% reporting that a critical incident occurred within the past month [14]. Similarly, 90% in a sample of firefighters reported at least one critical incident in the past year, with the average number of exposures being over six per year [15]. As a result of these high exposures, first responders may develop mental health disorders. First responders display symptoms of anxiety, depression, substance use disorders, post-traumatic and stress disorder [16], as well as suicidal ideation and suicide attempts [17] at rates higher than the general population. 

To address these high rates of mental health concerns, efforts have been developed and implemented to address the effects of trauma due to occupational exposure. Traditional efforts include critical incident debriefing [18], programs to promote healthy lifestyle [19], and clinical treatment that can take place through an EAP program or behavioral health units of their agency or municipality [20]. Britt and McFadden [21] describe a dilemma for first responder mental health treatment. That is, in many cases receiving treatment depends on treatment-seeking on the part of the first responder, yet a critical barrier exists that discourages treatment-seeking behavior in first responder populations. These occupations tend to value mental toughness and first responders have historically perceived a stigma associated with seeking treatment for mental health concerns [21]. When first responders perceive stigma associated with treatment seeking, they are less likely to initiate care and continue to experience distress.

## 4. Stigma as a Barrier to Treatment-Seeking

Rates of treatment-seeking behaviors among first responders experiencing mental health concerns are low. Sixty percent of first responders in a large representative sample reported a need for care, but among those reporting a need, less than half reported treatment-seeking behavior [22]. Although more research is needed on the topic of service utilization, existing literature suggests that first responders may expect or actually experience negative consequences as a result of treatment-seeking [4,23,24]. Although some other barriers do exist (i.e., lacking time to seek treatment or lack of trust in providers), perceived or experienced stigma is often reported as a reason for not utilizing services, such as an EAP [4]. That is, first responders who do not utilize these services when there is a need report not wanting to appear weak, not wanting to be treated differently by peers and leaders, and not wanting to experience career repercussions [4,21,23,24,25,26]. 

Perceived or experienced stigma for treatment-seeking in first responder populations originate from cues within the environment about expected behavior, with many messages espousing the idea that if the responder cannot “just get over it,” that is a sign of weakness [27,28]. When asking police officers what they think most people believe, 45.9% reported that most people would see being treated for a mental illness as a sign of personal failure [29]. This perception has negative consequences for treatment-seeking, as in the same sample of police officers, 44.4% reported that most first responders would not seek professional help when appropriate.

## 5. Peer Support Interventions to Promote Treatment-Seeking

Clearly, interventions are needed to encourage treatment-seeking behavior in first responders when appropriate. Although workplace interventions can vary in their targets and strategies [30], one method that warrants further attention is to leverage work peers to promote healthy behaviors in an employee [31]. Peer-focused strategies can involve peer modeling, in which peers who already possess or demonstrate the target skill, behavior, and health status are paired with someone who is at risk or needs improvement in the target variable [32]. This intervention method leverages observational learning theory [33] to motivate self-improvement through upward comparison. That is, the employee is inspired by their healthy peers. Other peer-focused interventions could leverage more direct forms of peer involvement. For example, most peer-based interventions feature components of instrumental and emotional support [34]. Although this strategy has been applied in community settings, it is currently underutilized in workplaces. This is surprising, given the strong influence of work peers on employee health [35].

Peer-focused interventions could be particularly useful in this context given that social norms within close social networks, including norms among co-workers, influence perceptions of stigma and ultimately treatment-seeking behavior [36]. A peer-focused intervention could promote perceptions that treatment-seeking behavior is encouraged when needed, rather than a sign of weakness. Peer support is defined as “a system of giving and receiving help founded on key principles of respect, shared responsibility, and mutual agreement of what is helpful” and these programs are not specific to a single condition or setting [37]. Peer support interventions are efficacious as a method of secondary prevention [38,39] and can be applied in a variety of occupations or settings. For example, peer support interventions have been used with success as a method of first-line psychological intervention in high-risk groups such as military personnel [40,41] and firefighters [6]. 

Evidence suggests that peer-focused interventions may be preferred to other forms of interventions in high-risk occupations for several reasons. First, work peers can better understand the features of the job, which promotes the expression of genuine empathy for the intervening peer [39]. The fact that the supporter “really gets the job” and “has walked in their shoes” can also promote buy-in for the first responder with mental health concerns. Second, first responders tend to prefer informal post-incident intervention methods [42] and peers may achieve a relatable, informal tone over other sources. Finally, first responders tend to trust their peers more than mental health professionals [39]. In the context of mental health, peers are not providing therapy, but rather they provide support, normalize the use of services, and encourage and assist the first responder in accessing a higher level of care, such as an EAP [39]. 

Research shows that peer support programs are associated with increases in inclusion, team cohesion and success, and building social networks at work [39] and have important implications for both behavioral outcomes and safety outcomes [43]. This is particularly encouraging given that worry over social exclusion is such a salient barrier to treatment-seeking. That is, it seems that leveraging the social environment can not only be effective modifying individual behavior, but it may improve the social environment itself. However, there has been little attention in the literature on peer support interventions to how pre-existing conditions may influence the success of these interventions. 

## 6. Individual and Interpersonal Intervention Context

All workplace interventions, peer-focused or otherwise, take place against the backdrop of existing context, such as the pre-existing social environment. Qualities of the pre-existing social context are relevant to a workplace intervention for several reasons. First, the social environment can influence the target of a workplace intervention. For example, interventions seek to modify health behavior or safety behavior, which are shaped by social norms [44,45], or health and safety outcomes, which are influenced by social relationships [46]. Second, an intervention could use social tactics to achieve desired health and safety outcomes. Such interventions are based on socio-cognitive foundations and acknowledge the social determinants of health and safety [47]. Finally, the social environment can represent a confounding factor that can either help or hurt intervention implementation [48]. Few intervention studies directly measure implementation factors that can influence outcomes, but among those that do, notable social factors are present. These include perceived motivation for offering the intervention, employee support, and leader support [48]. Recent examinations of low uptake of peer-directed social support interventions in high-risk occupations supports the notion of preexisting social environments being an important process variable. Jessiman-Perreault and colleagues [48] adopt a theoretical approach to demonstrate how features of the job itself, structural conditions, and pre-existing social capital among peers will influence intervention success for better or worse.

There would be benefits to more frequent consideration of the preexisting context surrounding an intervention. It is well situated within the best practice of process evaluation, which examines when and why an intervention achieves desired outcomes [49]. For example, a widely used model of process evaluation for organizational interventions recommends measuring and accounting for shared mental models among groups of participants [50], highlighting the importance of social processes in promoting the success or failure of an intervention. It is possible that existing individual differences or interpersonal perceptions may create a floor effect in an intervention, wherein a participant must possess a certain level of the dependent variable or a contextual variable in order to benefit from the intervention. There have been calls for more research on ceiling and floor effects in interventions in occupational health psychology [51], given that pre-intervention circumstances may promote or hinder intervention success for certain groups. These circumstances could include pre-intervention levels of the target variable. Nielsen and colleagues [51] discuss that organizations may need a certain level of “healthiness or readiness” for an intervention to be successful (i.e., a floor effect). The same could be true of an individual intervention participant. A pre-existing level of efficacy or related personal resource could serve as a foundation for the intervention to thrive. 

## 7. The Present Study

As peer-focused interventions are growing in popularity [52], there is a need to better understand contextual factors that will lend to the success of these interventions in a variety of occupational contexts. This is especially true in high-risk occupational settings, where both mental health burden and treatment-seeking stigma are high. This paper will first present an outcome evaluation, replicating previous research demonstrating that peer mental health support interventions are effective in increasing self-efficacy to have supportive conversations with a first responder peer in distress. 

**Hypothesis** **1** **(H1).**
*There will be a significant increase in (a) intention to use and (b) self-efficacy to use intervention concepts from pre-intervention to post-intervention.*


We will also consider the pre-intervention levels of the target variable (e.g., pre-intervention self-efficacy) and the preexisting workplace social environment as an intervention process variable, examining their influence on the outcomes of a peer support intervention for first responders. Although several variables exist that capture the quality of social relationships in the workplace, trust has been named in intervention process models as an important contextual factor that can help or hinder intervention implementation [53]. Trust is defined as “the willingness of a party to be vulnerable to the actions of another party based on the expectation that the other will perform a particular action important to the trustor, irrespective of the ability to monitor or control that other party” [54]. High levels of trust in peers would likely increase intervention participants’ confidence in the “other actor” (their peer) in a conversation about mental health, as they may be more sure that their experiences will remain confidential and that their peer’s reassurance and guidance is well-intentioned.

**Hypothesis** **2** **(H2).**
*Pre-intervention levels of intention and interpersonal trust in peers will influence post-intervention levels of intention to use intervention concepts.*


**Hypothesis** **3** **(H3).**
*Pre-intervention levels of self-efficacy and interpersonal trust in peers will influence post-intervention levels of self-efficacy to use intervention concepts.*


## 8. Methods

### 8.1. Participants

Participants were first responders in the Central Florida community who participated in a peer mental health support paraprofessional training. According to a trainee census maintained by the training organization, 280 first responders completed the training from March 2019 to February 2020. Among those trainees, 197 first responders completed pre-intervention and post-intervention surveys, earning a response rate of 70%. Due to concerns over the sensitive nature of data collected, demographic characteristics were not assessed in the survey. However, data from the trainee census (which is not connected to survey responses) revealed that the entire pool of intervention trainees were primarily male (71.42%) and primarily employed in fire (58.93%) and police services (30.36%). Although data are nested within agency, ICCs (−0.03 for post-intervention self-efficacy; −0.09 for post-intervention intention;) revealed that the majority of variance exists at the within-department level rather than the between-department level, which warrants an analysis without a multilevel component.

### 8.2. Procedure

All research activities were approved by the University of Central Florida Institutional Review Board. Trainees completed an eight-hour, group-based, in-person peer mental health support intervention. REACT (Recognize, Evaluate, Advocate, Coordinate, and Track) is a paraprofessional program designed to train the first responder stakeholders to deliver peer support. REACT was developed in partnership with public safety agencies to address the need for promoting psychological health among first responders and facilitated by licensed clinical psychologists and doctoral students in clinical psychology. Details regarding intervention development, implementation, and pilot evidence for efficacy are documented by Marks and colleagues [6] and a short description of training procedures are described below. 

Training content included psychoeducation regarding mental health, including content specifically related to mental health in first responders. This portion of the trainees introduced the idea of critical incidents as having the potential to produce mental health distress (referred to as stress injuries in the training). Trainees were then instructed on the recognition and evaluation of severity of behavioral health indicators of stress injuries (e.g., substance abuse, anger management issues, performance or attendance issues, etc.). They were given communication tools from motivational interviewing to facilitate a supportive conversation and to encourage their peer to seek a higher level of care. These tools included asking open-ended questions, providing validating and empathetic responses, and directly asking about intent to harm self or others when applicable to the severity of the behavioral health problem. Following didactic content, trainees participated in role-play exercises designed to reinforce these intervention targets and increase self-efficacy to provide peer support. A doctoral student or faculty member in Clinical Psychology observed the role-play scenarios and provided real-time feedback to the first responders. In some sessions, all trainees belonged to a single agency, while in others multiple agencies were present for a single training.

The research design was a quasi-experimental one-group pre-test post-test design. At the beginning of the training, the clinicians explained that trainees were invited to participate in a voluntary research study about factors that could influence their experience in the training. Interested participants provided informed consent and completed the pre-intervention survey, either on paper or on their mobile device depending on their preferences. Immediately after the training, trainees were given the opportunity to complete the post-intervention survey in a similar manner. Responses were anonymous and pre- and post-intervention surveys were linked using a participant-generated ID code. As mentioned above, 197 matched pre- and post-test surveys were completed. Reasons for non-response could include a lack of interest in the research study, only completing one out of the two surveys, or misremembering their ID code, preventing the matching of surveys. 

### 8.3. Measures

**Interpersonal trust.** Interpersonal trust in peers was included in the pre-intervention survey as an intervention process variable. Three items by Cook and Wall [55] were included (α = 0.83). Initial work with this scale provided evidence for reliability, the hypothesized factor structure, and construct validity [55]. The items are: “I can trust the people I work with to lend me a hand when I need it,” “Most of my workmates can be relied upon to do as they say they will do,” and “I have full confidence in the skills of my workmates.” Participants rated their level of agreement with each statement on a Likert scale from one to five (“strongly disagree” to “strongly agree”). Responses were averaged with higher scores representing higher levels of trust in peers. General interpersonal trust was measured, as opposed to trust related specifically to mental health concerns. 

**Self-efficacy to use intervention concepts.** Self-efficacy to use intervention concepts was measured at pre-intervention and post-intervention as an intervention outcome variable. REACT has been previously evaluated using this outcome variable [6]. The 25-item scale was created based on a published guide for creating self-efficacy scales [56] (pre-intervention α = 0.97; post-test α = 0.97). The item stems referred to the behavioral targets of the intervention (e.g., “I can motivate a peer to seek a higher level of care”). Given that scales constructed using this guide will have unique reference points (i.e., another researcher could have used this guide to create a “self-efficacy to detach from work” scale, while we created a “self-efficacy to use intervention concepts to communicate with peers regarding mental health” scale), there is no single validation of this exact scale that we can cite. However, the guide contains comprehensive recommendations regarding content, predictive, construct, and face validity [56]. Participants were instructed to rate their level of certainty that they could perform the action on a scale from 0 (“cannot do at all”) to 100 (“highly certain I can do”). Similar to Marks and colleagues [6], participants completed a practice self-efficacy scale in which they rated their certainty that they could complete a familiar task, lifting a weight of increasing difficulty. Responses on the self-efficacy scale were averaged, with higher scores representing higher self-efficacy to perform intervention targets in the future. 

**Intention to use intervention concepts.** As confidence to perform an action and intentions to perform an action are related, yet distinct concepts (and both important for behavioral change in an intervention context [57]), intention to use intervention concepts was measured pre- and post-intervention as an intervention outcome variable. The five-item scale that was constructed based on the intervention learning objectives (pre-intervention α = 0.76; post-intervention α = 0.89). An example item includes “I intend to use open-ended questions when talking to my coworkers”). Participants rated their likelihood to perform the action on a scale from one (“very unlikely”) to five (“very likely”). Responses were averaged with higher scores representing a stronger intention to perform intervention targets in the future.

## 9. Results

Descriptive statistics and correlations among study variables can be found in Table 1. To test Hypothesis One, paired samples t-tests were conducted comparing intention and self-efficacy at pre- and post-intervention. Results revealed that both intention, t(166) = 15.15, *p* < 0.01, and self-efficacy, t(161) = 23.30, *p* < 0.01, significantly increased following the intervention. Thus, Hypothesis 1a,b were supported.

To test Hypotheses Two and Three, median splits were calculated to create the following categorical variables: pre-intervention intention (low and high), pre-intervention self-efficacy (low and high), and interpersonal trust in peers (low and high). Means in dependent variables among each group can be found in Table 2. A univariate ANCOVA was conducted in the low and high pre-intervention intention groups that specified post-intervention intention as a dependent variable, pre-intervention intention as a control variable, and the categorical interpersonal trust in peers variable as a fixed factor. The same analysis was repeated for self-efficacy. Such an analysis examines whether or not group membership into high and low interpersonal trust in peer groups predicts post-intervention scores after controlling for pre-intervention scores. We first tested to ensure that the data met assumptions necessary to perform an ANOVA-based analytical approach; the dependent variables were continuous, independent variables were categorical groups, observations were independent, and there were no meaningful outliers. Although Shapiro–Wilke tests revealed that dependent variables were not normally distributed, F and T family tests are typically robust against this violation as long as the sample size exceeds 30. Levene’s Test of Homogeneity of Variances revealed that variances were equal among the median split groups for the fixed factor, interpersonal trust in peers (F(1163) = 2.70, *p* = 0.10).

Results for intention to use intervention concepts revealed that in the group with low pre-intervention intention levels, membership to high and low interpersonal trust in peers groups was not related post-intervention intention levels, F(1,77) = 0.51, *p* = 0.48. The same was true in the group with high pre-intervention intention levels, F(1,82) = 0.15, *p* = 0.70. Thus, Hypothesis Two was not supported.

Results for self-efficacy to use intervention concepts revealed that in the group with low pre-intervention self-efficacy levels, membership to high and low interpersonal trust in peers groups was not related to post-intervention self-efficacy levels, F(1,90) = 0.11, *p* = 0.74. However, in the group with high pre-intervention self-efficacy levels, membership to high and low interpersonal trust in peer groups was related post-intervention self-efficacy levels, F(1,64) = 4.43, *p* < 0.05. Post-intervention means were higher in the group with high levels of interpersonal trust in peers (M = 11.54), compared to the group with low levels of interpersonal trust in peers (M = 10.84). Thus, Hypothesis Three was supported.

## 10. Discussion

In this study, we first sought to replicate previous research supporting the peer mental health support interventions in first responder populations. This is particularly important given the high burden associated with cumulative stress from critical incidents in first responder populations, high rates of mental health distress and associated behavioral health issues, and high rates of suicide. Consistent with previous research, the results suggest that a peer mental health support intervention may be efficacious in producing changes in self-efficacy and intention to communicate with peers regarding mental health, directly assess intent to harm self or others, and to encourage treatment seeking in peers in first responder populations.

We also sought to extend previous research on peer mental health support interventions by examining how pre-existing contextual variables could influence the intervention process. That is, this peer-focused intervention is conducted in a group format and target changes that leverage interpersonal interactions, such as lending a supportive ear to a peer in need. It stands to reason that the intervention is implemented within a historical individual and social context that could help or hinder the intervention. In this study, we examined how pre-existing levels of intervention target variables (self-efficacy and intentions) and pre-existing levels of interpersonal trust relate to intervention outcomes. Results provide initial insight into the possible importance of attention to pre-existing context surrounding a peer-focused intervention; although differences are not observed for intention to use intervention concepts, baseline levels of the target variable and contextual supports may influence how confident a first responder feels in using intervention concepts. Post-intervention self-efficacy levels are significantly greater when high levels of pre-intervention self-efficacy are accompanied by high levels of trust in peers.

The use of peer-focused intervention strategies and a focus on the social work environment are timely for several reasons. In light of several trends related to the future of work, these practices are likely needed to address the increased importance of solutions based in organizational design to address work-related stresses, the need to collaboratively engage employees and leaders in safety and health approaches, and the value of fostering trust between employees and organizations to fulfill corporate ethical responsibilities [8]. These results lend support to the notion that trust in organizations (or parties representative of the organization) matter and that this trust could influence the ways in which organizational design features to support employees (such as EAPs) are used or not used. As the workplace continues to change and complexities arise in work, workplaces, and workers, context will continue to prove important as we ensure that valuable interventions are not implemented in environments in which they would not thrive.

### 10.1. Implications for Research and Practice

These results bear a number of implications for research. If using a general framework of process evaluation as examinations of when and why an intervention works, these results bolster previous findings that it is important to consider both process and outcomes in intervention evaluation. Although not comparable to more comprehensive and rigorous process evaluation models, the findings of this study may shed light on the importance of process evaluation models that contain categories of social contextual variables. Although discussions of floor effects in this study are tempered based on the limitations of the study design, the results could suggest the importance of further research on floor or ceiling effects in intervention implementation. Interventions are rarely delivered to a “blank canvas;” instead the intervention is interpreted in light of existing psychosocial context. Both discussions of floor and ceiling effects and guidance on how to detect them are in their infancy in occupational health psychology, and more attention to this phenomenon could help our field promote positive intervention outcomes for all groups of participants. 

These results also have important implications for practice. These results add to a growing body of evidence that peer-focused programming can protect safety and health in the workplace, even in occupations with a history of stigma surrounding vulnerable topics (as evidenced by the use of a peer-focused strategy to improve mental health treatment-seeking in first responders). These results suggest that the effectiveness of these strategies could be helped or hurt by the preexisting social environment in the agency. Practitioners could consider addressing social issues that would negatively influence trust before the implementation of a peer-focused intervention, to ensure that the environment supports the intervention. Efforts to ready an environment for an intervention, such as improving social capital prior to intervention implementation, could build contextual influences support rather than subdue an intervention.

### 10.2. Strengths, Limitations, and Future Research Directions

This study is strengthened by its focus on a timely topic of great significance, mental health distress in first responder populations and methods to reduce stigma and foster treatment seeking. It is also strengthened by its attention to not only the outcomes of an intervention, but the way context helps or hinders the intervention. This is based on the logic of process evaluation methodology, which is considered a best practice in intervention evaluation [58]. This study builds upon previous intervention work that examines interpersonal trust as context by examining trust along with individual difference variables, pre-existing levels of the intervention’s target variable. 

Despite these strengths, there are several limitations that should be noted. First, the evaluation methods relied on self-report measures and did not include behavioral outcome variables. Although intentions are an important predictor of behavior [59], future research should examine how social context influences behavioral outcome variables (such as actual use of intervention concepts over time) or objective outcomes variables (such as employee assistant program utilization rates for participating agencies). Additionally, it is a limitation that we were unable to control for department affiliation, even considering the low ICC values, given that agency is an important aspect of context. Finally, the study design limits the nature of conclusions that can be made from this study. The lack of a control group and random assignment prohibit causal conclusions about intervention effectiveness or the influence of contextual variables, and the lack of follow-up measures limit our interpretations about the sustained success of the intervention. 

Future research should examine intervention outcomes over a longer period of follow-up and randomization into a control group. Not only is this necessary to support causal claims about intervention efficacy, but this would also allow researchers to assess the influence of trust with more sophisticated statistical approaches (for example, based on a helpful reviewer’s suggestion, a randomized experiment would allow for the intervention to serve as an independent variable (treatment vs. control or varying levels of the intervention) and trust could be treated as a moderator). We also note that the use of median splits is not the only method of creating groups based on pre-intervention levels of various constructs. We chose a median split to mimic the comparison of two groups in earlier works that discuss ceiling and floor effects [51]. However, future research should aim to identify the most effective methods of group comparison in examinations of floor and ceiling effects. In addition, the sample of this study is relatively small and due to the anonymous nature of data collection, we were unable to examine how social processes or intervention process and outcomes might vary as a function of sociodemographic variables. Finally, this study focuses on how social context influences an intervention, but it could be possible that an intervention could in turn modify the social environment. Interpersonal trust was only measured in the pre-intervention survey because we found it unlikely that intervention-induced social change would permeate an agency immediately after a single-day intervention. We found it more likely that over time participants would adhere to the intervention’s behavioral recommendations in interpersonal interactions, shaping trust over time after the intervention. Future research could investigate the potential reciprocal relationship between intervention and environment, by including interpersonal trust in a follow-up survey that occurs weeks or months after the intervention. 

### 10.3. Conclusions

Peer-focused interventions have been used with some success to improve occupational safety, health, and well-being. Peer support interventions have been used to promote treatment-seeking for mental health among first responders, a population that struggles with high levels of depression, PTSD, and suicide paired with high levels of mental health stigma. However, research has not yet accounted for how the pre-existing context may influence the success of these interventions. We found evidence for the efficacy of the peer support interventions to increase self-efficacy and intention to communicate with peers regarding mental health, directly assess intent to harm self or others, and to encourage treatment seeking in first responders. We also found that pre-existing conditions matter; self-efficacy to use intervention concepts was highest when pre-existing self-efficacy and trust in peers was high. These results support an increased consideration of individual and interpersonal context prior to and during an intervention in order to maximize intervention success.

## Figures and Tables

**Table 1 ijerph-18-11097-t001:** Descriptive Statistics and Correlations.

	**Mean**	**SD**	**(1)**	**(2)**	**(3)**	**(4)**	**(5)**
1. Peer Trust	4.04	0.67	(0.83)				
2. Pre SE	6.64	2.42	0.12	(0.97)			
3. Post SE	10.66	1.91	0.16 *	0.41 **	(0.97)		
4. Pre Int	3.81	0.60	0.12	0.42 **	0.26 **	(0.76)	
5. Post Int	4.69	0.56	0.03	0.03	0.17 *	0.25	(0.89)

Note: N = 197 first responders who completed pre and post surveys for a peer mental health support intervention; SE refers to self-efficacy and Int refers to intentions; * indicates *p* < 0.05, ** indicates *p* < 0.01.

**Table 2 ijerph-18-11097-t002:** Means Among Median Split Groups.

	Dependent Variable: Post SE		Dependent Variable: Post Int	
	Low Pre SE	High Pre SE	Low Pre Int	High Pre Int
Low Peer Trust	10.53 (N = 31)	10.85 * (N = 14)	4.55 (N = 24)	4.83 (N = 21)
High Peer Trust	10.37 (N = 62)	11.54 * (N = 53)	4.64 (N = 56)	4.78 (N = 64)

Note: N = 197 first responders who completed pre and post surveys for a peer mental health support intervention; SE refers to self-efficacy and Int refers to intentions; * indicates that groups were significantly different based on ANCOVA.

## Data Availability

The data presented in this study are available on request from the corresponding author. The data are not publicly available due to the sensitive nature of mental health topics and job attitudes collected in the survey.

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
