# Peer review of "Here for My Peer: The Future of First Responder Mental Health"

_ijerph, 2021, doi:10.3390/ijerph182111097_

Round 1

Reviewer 1 Report

The idea is greate. Congratulations. However, there is a gap which concerns the relation with sociodemographical variables. In this context the research group is to small. 

Author Response

Reviewer One:

Reviewer One, Comment One: The idea is great. Congratulations. However, there is a gap which concerns the relation with sociodemographic variables. In this context the research group is too small.

            Response to Reviewer One, Comment One: We appreciate your encouraging comments regarding the manuscript! The point about sociodemographic variables is well-taken. We have added this text to the discussion section:

“In addition, the sample of this study is relatively small and due to the anonymous nature of data collection we were unable to examine how social processes or intervention process and outcomes might vary as a function of sociodemographic variables.”

Reviewer 2 Report

This is an interesting study, and no doubt provides important information on context when implementing interventions. 

Some comments:

  1. Please justify using median splits (as opposed to other ideas like tertiles?).
  2. Please be clear on the design of your study: Quasi experimental one group pre-test post test design? 
  3.  Please add more limitations of this design - there are many as there is no control/comparison group.
  4. The discussion is good, but does not integrate much previous research (make stronger links back to the literature review).
  5. Can you please check that the ANOVA is not a specific type of ANOVA that should be named as such (e.g. ANCOVA, ANCOVA-CHANGE, ANOVA-CHANGE).  I ran out of time to research this myself. Please check if post hoc analyses would be useful to report. 

Author Response

Reviewer Two, Comment One: This is an interesting study, and no doubt provides important information on context when implementing interventions. 

            Response to Reviewer Two, Comment One: Thank you for your positive comments!

Reviewer Two, Comment Two: Please justify using median splits (as opposed to other ideas like tertiles?).

            Response to Reviewer Two, Comment Two: Thank you for this helpful suggestion. The creation of two groups was based off of a comparison of two groups (Group A and Group B) in Nielsen et al. (2006) discussion of ceiling effects.  However, their groups were two intervention sites, rather than groups created based on scores. To address this comment, we have added the following text:

“We also note that the use of median splits is not the only method of creating groups based on pre-intervention levels of various constructs. We chose a median split to mimic the comparison of two groups in earlier works that discuss ceiling and floor effects [52]. However, future research should aim to identify the most effective methods of group comparison in examinations of floor and ceiling effects.”

Reviewer Two, Comment Three: Please be clear on the design of your study: Quasi experimental one group pre-test post test design? 

            Response to Reviewer Two, Comment Three: We have added a sentence to the methods section identifying the design of the study.

Reviewer Two, Comment Four: Please add more limitations of this design - there are many as there is no control/comparison group.

Response to Reviewer Two, Comment Four: This is an important point. To better represent the conclusions that can be made from this study, we have added the following text to the discussion section:

 “Importantly, the design of the study is associated with several limitations; we cannot conclude that the peer mental health support intervention leads to greater efficacy and intentions to communicate with peers regarding mental health. A randomized experiment would be necessary to support those claims.”

Reviewer Two, Comment Five: The discussion is good, but does not integrate much previous research (make stronger links back to the literature review).

            Response to Reviewer Two, Comment Five: In light of this helpful comment, we have added content to the first paragraph of the discussion that more clearly relates to the introduction.

Reviewer Two, Comment Six: Can you please check that the ANOVA is not a specific type of ANOVA that should be named as such (e.g. ANCOVA, ANCOVA-CHANGE, ANOVA-CHANGE).  I ran out of time to research this myself. Please check if post hoc analyses would be useful to report. 

            Response to Reviewer Two, Comment Six: Thank you for this helpful suggestion. We have updated the label ANOVA to ANCOVA based on the presence of a covariate in the analysis.

Reviewer 3 Report

In the manuscript entitled “Here for my Peer: The Future of First Responder Mental Health,” the authors examined the effect of a peer mental health support interventions for first responder population. I appreciate that the authors tackle an important topic, but the design and data analysis should be revised before it can be published. Below I lay out major comments, followed by minor and editorial ones:

Major comment:

  1. The authors should be clear about what kind of trust they are referring to. Being one of the critical factors in the study, it seems that the authors discussed the trust of receiving support from peers, or at least keeping it a secret when a first responder confides in a peer. However, the measure of trust, at least based on the sample item, reads like a trust in carrying out their daily job (“Most of my workmates can be relied upon to do as they say they will do”). If that is the case, please review relevant literature on how such trust is related to mental health.
  2. Related to the previous comment, the authors should reconsider the role of trust in the statistical model. Trust was only measured before the intervention and was entered into ANOVA as an independent variable. The authors should consider treating trust as a moderator because 1) trust was not part of the intervention and 2) trust was conceptualized as a contextual factor (section 6). In the lit review, it seems that trust shapes the effect of the training: the training will be more successful, however it is defined, in an environment where the perceived trust is high. A better alignment between the role of trust and statistical models should be in place.
  3. I suggest that the authors revamp the statistical analyses and revised the phrasing in the Results, Discussion, and Implications section. In addition to the moderation effect mentioned above, the authors should not split the variables into high-low groups but keep them continuous. Otherwise, the authors should report whether the assumptions of t-tests and ANOVA hold, particularly the homoscedasticity assumption. Second, it is difficult to see the floor effects based on the current data analyses. Non-significant difference does not mean the existence of floor effects. The authors should at least report descriptive statistics to demonstrate such effect. Third, please tone down in the Results, Discussion, and Implications sections because of a lack of control group. Based on the authors’ background in psychology, I am sure the authors are very clear about how weak the causal evidence is given the study design. I understand that a random assignment and having a control group is not always feasible in field work, but the authors can be more cautious when interpreting the results and discuss this weakness in the Limitations section. For example, line 387-388 should be toned down, because, honestly, the design is weak; it is not really a process evaluation methodology, but a simple pre- and post-comparison with no control group, no random assignment, and no information about the influence of attrition (i.e., fidelity). It is OK, we do not live in a perfect world, but please be truthful and candidly discuss the limitations. Please thoroughly revise the manuscript.

Minor comments:

  1. Please acknowledge the limitation of the inability to control for the department affiliation despite the low ICCs. This paper is on the contextual influences, so this important contextual information should be addressed. I understand that the limitation is beyond the authors’ control, so please discuss this limitation properly.
  2. (line 287-292) Please include all three items since it is a short scale. Please also mention how the items are rated (e.g., 5-point Likert-type scale). Please also state why interpersonal trust was not measured again after the intervention, or please discuss its limitation when not included.
  3. (7.3 Measures) Please report the validity information for all scales.

Editorial:

  1. (line 55) Spell out EAP.
  2. (line 244) “-.03” missing a decimal point
  3. (line 249) Please specify the institution that granted IRB approval. It is in the Institutional Review Board Statement anyway, so the same information should be presented in the main text as well.
  4. Please proofread the manuscript, including the references (e.g., #57 does not seem correct).

Author Response

Reviewer Three, Comment One: In the manuscript entitled “Here for my Peer: The Future of First Responder Mental Health,” the authors examined the effect of a peer mental health support interventions for first responder population. I appreciate that the authors tackle an important topic, but the design and data analysis should be revised before it can be published. Below I lay out major comments, followed by minor and editorial ones.

            Response to Reviewer Three, Comment One: Thank you for your comments and helpful suggestions.

Reviewer Three, Comment Two: The authors should be clear about what kind of trust they are referring to. Being one of the critical factors in the study, it seems that the authors discussed the trust of receiving support from peers, or at least keeping it a secret when a first responder confides in a peer. However, the measure of trust, at least based on the sample item, reads like a trust in carrying out their daily job (“Most of my workmates can be relied upon to do as they say they will do”). If that is the case, please review relevant literature on how such trust is related to mental health.

            Response to Reviewer Three, Comment Two: This is an important point. We assessed general workplace trust, with measures that correspond to this broad conceptualization. We assert that your general level of trust in your peer will influence your intentions to confide in your peer about mental health concern. During intervention sessions, we heard many mentions to peers “having your back.” This was discussed in terms of both task performance (performing protective peer-oriented behaviors during dangerous calls) as well as social contextual performance (lending a supportive ear if a first responder opened up about mental health concerns). In order to avoid confusion, we have added a sentence to the measures section that identifies the type of trust measured.

Reviewer Three, Comment Three: Related to the previous comment, the authors should reconsider the role of trust in the statistical model. Trust was only measured before the intervention and was entered into ANOVA as an independent variable. The authors should consider treating trust as a moderator because 1) trust was not part of the intervention and 2) trust was conceptualized as a contextual factor (section 6). In the lit review, it seems that trust shapes the effect of the training: the training will be more successful, however it is defined, in an environment where the perceived trust is high. A better alignment between the role of trust and statistical models should be in place.

            Response to Reviewer Three, Comment Three: We appreciate this helpful comment. After careful consideration, we still believe that an ANCOVA is appropriate based on our research question. We assert that pre-existing levels of trust would be helpful in any socially-based intervention, regardless of if the intervention targeted trust or not. Additionally, in this case it would be impossible to perform a moderator analysis given that there is no control group (at least to our knowledge). That is, we cannot treat the intervention itself as an independent variable because there is no control group and no levels of the intervention. Based on these limitations, we chose to examine intervention outcomes after controlling for pre-intervention starting points in dependent variables. However, we wanted to ensure that your analytical suggestions were still represented in the manuscript in some way. We added content to the limitations and future research directions section that describes more sophisticated analyses that could be used with alternative research designs.

Reviewer Three, Comment Four: I suggest that the authors revamp the statistical analyses and revised the phrasing in the Results, Discussion, and Implications section. In addition to the moderation effect mentioned above, the authors should not split the variables into high-low groups but keep them continuous. Otherwise, the authors should report whether the assumptions of t-tests and ANOVA hold, particularly the homoscedasticity assumption.

            Response to Reviewer Three, Comment Four: We opted to retain the analytical approach based on the limitations associated with the design described in the previous comment.  We agree that checking assumptions associated with this approach is important. We have checked assumptions and included the results in the paragraph describing the analytical approach for hypothesis two and three.

Reviewer Three, Comment Five: Second, it is difficult to see the floor effects based on the current data analyses. Non-significant difference does not mean the existence of floor effects. The authors should at least report descriptive statistics to demonstrate such effect.

            Response to Reviewer Three, Comment Five: We based our interpretation of floor effects based on Nielsen et al. (2006)’s discussion of a floor effect. One intervention site reported significant intervention effects, while the other did not. The successful site had more favorable pre-intervention conditions, which led the authors to speculate that a worksite may have to achieve a certain “floor” of positive psychosocial conditions in order to reap the benefit of an intervention. Of course, our design is slightly different than Nielsen et al. (2006), as we created groups based on levels of a pre-intervention psychosocial conditions rather than observing differences across naturally occurring groups. But given that the practice examination of ceiling and flood effects in intervention contexts is in its infancy, we opted to retain the general logic of Nielsen et al. (2006)   If the reviewer is aware of any alternative criteria (beyond significant vs. nonsignificant effects) that are more appropriate for this design, we would be happy to investigate them further. Per your helpful suggestion, we have added two tables: descriptive statistics and correlations for the entire sample and means in dependent variables in the median split groups.

Reviewer Three, Comment Six: Third, please tone down in the Results, Discussion, and Implications sections because of a lack of control group. Based on the authors’ background in psychology, I am sure the authors are very clear about how weak the causal evidence is given the study design. I understand that a random assignment and having a control group is not always feasible in field work, but the authors can be more cautious when interpreting the results and discuss this weakness in the Limitations section. For example, line 387-388 should be toned down, because, honestly, the design is weak; it is not really a process evaluation methodology, but a simple pre- and post-comparison with no control group, no random assignment, and no information about the influence of attrition (i.e., fidelity). It is OK, we do not live in a perfect world, but please be truthful and candidly discuss the limitations. Please thoroughly revise the manuscript.

            Response to Reviewer Three, Comment Six: The reviewer is indeed correct about the tradeoffs that can exist between the constraints of field data collection and ideal study design, as was the case in this study! However, we completely agree on the importance of matching the tone of the conclusions to the strength of the design and apologize for the overstatement of results. In addition to generally toning down the results and discussion, we have specifically addressed the following statements.

  • In response to the concern about the label of process evaluation, we modified the text to say: “If using a general framework of process evaluation as examinations of when and why an intervention works, these results bolster previous findings that it is important to consider both process and outcomes in intervention evaluation. Although not comparable to more comprehensive and rigorous process evaluation models, the findings of this study may shed light on the importance of process evaluation models that contain categories of social contextual variables.”  
  • In response to the concern about the study design, we modified the text to say: “Additionally, the study design limits the nature of conclusions that can be made from this study. The lack of a control group and random assignment prohibit causal conclusions about intervention effectiveness or the influence of contextual variables, and the lack of follow-up measures limit our interpretations about the sustained success of the intervention. Future research should examine intervention outcomes over a longer period of follow-up and randomization into a control group. Not only is this necessary to support causal claims about intervention efficacy, but this would also allow researchers to assess the influence of trust with more sophisticated statistical approaches.”
  • On a minor note, we chose to not mention attrition as an indicator of fidelity for a couple of reasons: First, every single first responder experienced the whole intervention session regardless of if they participated in the survey or filled out both surveys. Although fidelity could vary across cohorts of participants (i.e. agencies receiving the single intervention session on different days), these differences are expected to be relatively small because the same clinical psychologists delivered all session, they followed a structured protocol for each intervention delivery, and participants filled out surveys immediately before and after the single intervention session (so it is not as if a participant who did not fill out the follow up survey missed one or more sessions in a multiple session intervention and missed out on intervention components). Second, the most common reasons for the inability to match pre and post-test surveys seemed to be accidental (i.e., participants forgot to write their unique ID code or wrote in the wrong unique ID code).

Reviewer Three, Comment Seven: Please acknowledge the limitation of the inability to control for the department affiliation despite the low ICCs. This paper is on the contextual influences, so this important contextual information should be addressed. I understand that the limitation is beyond the authors’ control, so please discuss this limitation properly.

            Response to Reviewer Three, Comment Seven: Thank you for this suggestion. We have incorporated it into the limitations section.

Reviewer Three, Comment Eight: Please include all three items since it is a short scale. Please also mention how the items are rated (e.g., 5-point Likert-type scale). Please also state why interpersonal trust was not measured again after the intervention, or please discuss its limitation when not included.

            Response to Responder Three, Comment Eight: We have added all three items and the response scale to the measures section. We added the following information to the future directions section: “Finally, this study focuses on how social context influences an intervention, but it could be possible that an intervention could in turn modify the social environment. Interpersonal trust was only measured in the pre-intervention survey because we found it unlikely that intervention-induced social change would permeate an agency immediately after a single-day intervention. We found it more likely that over time participants would adhere to the intervention’s behavioral recommendations in interpersonal interactions, shaping trust over time after the intervention. Future research could investigate the potential reciprocal relationship between intervention and environment, by including interpersonal trust in a follow-up survey that occurs weeks or months after the intervention.”

Reviewer Three, Comment Nine: Please report the validity information for all scales.

Response to Reviewer Three, Comment Nine: We have added information about the validity of the interpersonal trust and self-efficacy scale. The intentions scale was created based on the learning objectives of the intervention, and therefore there is no previous validation study that we can cite for this scale.

Reviewer Three, Editorial Comments: (line 55) Spell out EAP; (line 244) “-.03” missing a decimal point; (line 249) Please specify the institution that granted IRB approval. It is in the Institutional Review Board Statement anyway, so the same information should be presented in the main text as well; Please proofread the manuscript, including the references (e.g., #57 does not seem correct).

Response to Reviewer Three, Editorial Comments: Thank you for including these helpful comments. Each of them has been addressed in the manuscript. Specifically, the last comment that listed reference #57, it actually was correct. I am assuming you were referring to the word “optimising,” and I confirmed that the title of the article does use this British spelling of the word. However, I am glad that you pointed this reference out, because I was missing the journal volume and page numbers, which are now added.

Round 2

Reviewer 3 Report

I believe the authors have properly addressed my concerns in the revised manuscript. It is not fun when the Reviewer 3 looks like "the" Reviewer 2, and I appreciate their patience and effort.